# Involvement of Organic Anion Transporters in the Pharmacokinetics and Drug Interaction of Rosmarinic Acid

**DOI:** 10.3390/pharmaceutics13010083

**Published:** 2021-01-09

**Authors:** Yun Ju Kang, Chul Haeng Lee, Soo-Jin Park, Hye Suk Lee, Min-Koo Choi, Im-Sook Song

**Affiliations:** 1College of Pharmacy and Research Institute of Pharmaceutical Sciences, Kyungpook National University, Daegu 41566, Korea; yun-ju6895@nate.com; 2College of Pharmacy, Dankook University, Cheonan 31116, Korea; hang1130@naver.com; 3College of Korean Medicine, Daegu Haany University, Daegu 38610, Korea; sjp124@dhu.ac.kr; 4College of Pharmacy, The Catholic University of Korea, Bucheon 14662, Korea; sianalee@catholic.ac.kr; 5BK21 FOUR Community-Based Intelligent Novel Drug Discovery Education Unit, Kyungpook National University, Daegu 41566, Korea

**Keywords:** rosmarinic acid, organic anion transporter (OAT), pharmacokinetics, herb-drug interaction

## Abstract

We investigated the involvement of drug transporters in the pharmacokinetics of rosmarinic acid in rats as well as the transporter-mediated drug interaction potential of rosmarinic acid in HEK293 cells overexpressing clinically important solute carrier transporters and also in rats. Intravenously injected rosmarinic acid showed bi-exponential decay and unchanged rosmarinic acid was mainly eliminated by urinary excretion, suggesting the involvement of transporters in its renal excretion. Rosmarinic acid showed organic anion transporter (OAT)1-mediated active transport with a K_m_ of 26.5 μM and a V_max_ of 69.0 pmol/min in HEK293 cells overexpressing OAT1, and the plasma concentrations of rosmarinic acid were increased by the co-injection of probenecid because of decreased renal excretion due to OAT1 inhibition. Rosmarinic acid inhibited the transport activities of OAT1, OAT3, organic anion transporting polypeptide (OATP)1B1, and OATP1B3 with IC_50_ values of 60.6 μM, 1.52 μM, 74.8 μM, and 91.3 μM, respectively, and the inhibitory effect of rosmarinic acid on OAT3 transport activity caused an in vivo pharmacokinetic interaction with furosemide by inhibiting its renal excretion and by increasing its plasma concentration. In conclusion, OAT1 and OAT3 are the major transporters that may regulate the pharmacokinetic properties of rosmarinic acid and may cause herb-drug interactions with rosmarinic acid, although their clinical relevance awaits further evaluation.

## 1. Introduction

Rosmarinic acid (Figure 1) is an ester of caffeic acid and 3,4-dihydroxyphenyllactic acid and a major phytochemical found in various dietary and medicinal herbs such as rosemary (*Rosmarinus officinalis*), perilla (*Perilla frutescens*), basil (*Ocimum basilicum*), *Mellisa officinalis*, *Origanum vulgare*, *Salvia officinalis*, and *Satureja hortensis* [1]. It is widely used for dietary ingredients and has been reported to possess various biological activities including anti-oxidative, anti-inflammatory, anti-apoptotic, and anti-bacterial activities [2]. Based on these biological activities, rosmarinic acid was evaluated in animal experimental disease models including liver fibrosis [3], cancer [4,5], metabolic syndrome [6], Adriamycin-induced cardiotoxicity [7], and cisplatin-induced nephrotoxicity [8]. 

In addition to therapeutic effects in animals and cell systems, the pharmacokinetic characteristics of rosmarinic acid in rats and human has been investigated following oral administration of rosmarinic acid or rosmarinic acid-containing herbal extracts. Noguchi-shinohara et al. [9] reported the pharmacokinetics of rosmarinic acid in humans following the oral administration of *Melissa officinalis* extract (500 mg as rosmarinic acid) in the fed and fasted states. The area under the plasma concentration curve (AUC) of rosmarinic acid in the fed state was higher than the AUC in the fasted state because of increased intestinal absorption of rosmarinic acid [9]. Baba et al. reported the absorption, metabolism, and urinary excretion properties of rosmarinic acid following a single oral dose of *Perilla frutescens* extract (200 mg as rosmarinic acid) in healthy humans [10]. The plasma concentrations of the parent rosmarinic acid and conjugated rosmarinic acid in human subjects were reported about 20 nM and 1200 nM, respectively, at 1 h after oral intake of the *Perilla frutescens* extract. Glucuronide conjugates of rosmarinic acid and methylrosmarinic acid were identified as the major metabolites of rosmarinic acid in urine samples [10]. Kim et al., using human recombinant isozymes, reported that cytochrome P450 (CYP)1A2, CYP2C19, CYP2E1, and CYP3A4, and uridine diphosphate glucuronosyltransferase (UGT)1A1, UGT1A6, and UGT2B7 were involved in the metabolism of rosmarinic acid [1]. Rosmarinic acid showed poor water solubility (1 mg/mL) and low partition coefficient (Log Kow = 1.82) [11]. In addition, rosmarinic acid showed low permeability, and 0.03–0.06% of rosmarinic acid was absorbed in Caco-2 cells via paracellular pathway [12] and it underwent glucuronide metabolism during the absorption phase [2]. In the dose range of 12.5–50 mg/kg, the absolute bioavailability of rosmarinic acid was 0.9–1.7% in rats, and their AUC did not show dose proportionality [13]. Oral administration of rosmarinic acid (50 mg/kg) in combination with piperine (20, 40, 60, and 80 mg/kg) significantly increased the AUC of rosmarinic acid together with a significant decrease of rosmarinic acid glucuronide [14]. These results indicated that the enhanced oral bioavailability of rosmarinic acid might be linked to the inhibition of UGT by piperine [15]. 

Except for the contribution of UGT metabolism in the intestine and hepatocytes, the role of drug-metabolizing enzymes and transporters in the pharmacokinetics of rosmarinic acid is largely unknown. Rosmarinic acid inhibited the catalytic activities of CYP2C19, CYP2E1, UGT1A1, UGT1A6, and UGT2B7 with K_i_ values of 31.6 μM, 42.4 μM, 6.7 μM, 14.2 μM, and 15.1 μM, respectively [1]. Because of its low bioavailability, the plasma rosmarinic concentration was too low compared with the K_i_ value for CYP and UGT enzymes to cause clinically relevant drug interactions. However, the rosmarinic acid concentrations in the intestine following the oral administration of rosmarinic acid (200 mg) could be calculated as 252–1100 μM, which can plausibly induce drug interactions [1].

One of the most frequently used traditional Chinese medicines containing rosmarinic acid is Shenxiong glucose injection [16]. In China, it has been used for various cardiovascular diseases and the major active components were danshensu, protocatechuic aldehyde, rosmarinic acid, and ligustrazine [17]. A comparison of the pharmacokinetics of ligustrazine after intravenous injection of Shenxiong glucose injection (6 mg/kg as ligustrazine) with those following intravenous injection of the same dose of ligustrazine alone (6 mg/kg), the AUC of ligustrazine after Shenxiong glucose injection was significantly higher than that after single ligustrazine injection. In addition, the AUC and clearance of rosmarinic acid did not show dose proportionality with its increasing dose (0.107 mg/kg, 0.16 mg/kg, and 0.32 mg/kg as rosmarinic acid) [17]. The results suggested the possibility of drug interactions among the major components of herbal drugs. In addition, intravenously injected rosmarinic acid showed significantly higher (more than 100-fold) kidney distribution compared with the heart, liver, spleen, lung, and brain [18]. The results suggested the involvement of drug transporters that are expressed mainly in the kidney in the pharmacokinetics and tissue distribution. Drug transporters, alone or in combination with drug-metabolizing enzymes, have drawn much attention in herb-drug interactions because they regulate the absorption, distribution, and elimination of drugs [19,20]. Therefore, the objectives of this study were to investigate the involvement of drug transporters in the pharmacokinetics of rosmarinic acid and to investigate the transporter-mediated drug interaction potential of rosmarinic acid. In this study, we included solute carrier transporters such as organic anion transporter (OAT)1, OAT3, organic anion transporting polypeptide (OATP)1B1, OATP1B3, organic cation transporter (OCT)1, OCT2, and Na^+^/taurocholate cotransporting polypeptide (NTCP) that are clinically important in terms of drug–drug interactions and herb–drug interaction [20,21].

## 2. Materials and Methods 

### 2.1. Materials 

Rosmarinic acid, probenecid, furosemide, valsartan, tetraethylammonium chloride, rifampin, cyclosporin A, Hank’s balanced salt solution (HBSS), sodium butyrate, and non-essential amino acids were purchased from Sigma-Aldrich (St. Louis, MO, USA). [^3^H]Para-aminohippuric acid (0.13 TBq/mmol), [^3^H]estrone-3-sulfate (2.12 TBq/mmol), [^3^H]estradiol-17β-D-glucuronide (2.22 TBq/mmol), [^3^H]taurocholate (0.57 TBq/mmol), and [^3^H]methyl-4-phenylpyridinium (2.9 TBq/mmol) were purchased from Perkin Elmer Inc. (Boston, MA, USA). Dulbecco’s modified Eagle’s medium (DMEM), fetal bovine serum (FBS), poly-D-lysine-coated 24-well plates, and poly-D-lysine-coated 96-well plates were purchased from Corning-Gentest (Tewksbury, MA, USA). Other chemicals were of the highest quality available.

HEK293-OAT1, -OAT3, -OATP1B1, -OATP1B3, -OCT1, -OCT2, and NTCP cells (HEK293 cells transiently overexpressing OAT1, OAT3, OATP1B1, OATP1B3, OCT1, OCT2, and NTCP transporters, respectively) and HEK293-mock cells were purchased from Corning Life Sciences (Woburn, MA, USA). 

### 2.2. Animals and Ethical Approval 

Male Sprague–Dawley rats (7–8 weeks old, 230–270 g) were purchased from the Samtako bio Korea, Inc. (Osan, Korea). Animals were acclimatized for 1 week in an animal facility at College of Pharmacy, Kyungpook National University, and food and water were supplied ad libitum. 

### 2.3. Pharmacokinetics of Rosmarinic Acid

The femoral artery and femoral vein were cannulated with polyethylene tubes (PE-50; Jungdo, Seoul, Korea) while the rats were anesthetized with Zoletil and Rompun (50 mg/kg and 5 mg/kg, respectively, intramuscular injection), and heparinized saline (10 U/mL) was used to prevent blood clotting. Rosmarinic acid was dissolved in saline and injected at doses of 1 and 10 mg/kg via the femoral vein (vehicle dosing volume, 1 mL/kg) after the recovery from the anesthesia. Blood samples (approximately 150 µL each) were collected from the femoral artery at 0, 0.033, 0.083, 0.167, 0.25, 0.5, 1, 1.5, 2, 4, and 6 h. After centrifugation of blood samples at 10,000× *g* for 1 min, plasma samples (25 µL) were collected and stored at −80 °C until liquid chromatography-tandem mass spectroscopy (LC-MS/MS) analysis. Urine and feces samples were collected for 24 h. Urine samples were diluted with 9-fold volumes of water and aliquots (25 µL each) were stored at −80 °C until LC-MS/MS analysis. Feces samples were homogenized with 9-fold volumes of water and aliquots (100 mg each) were stored at −80 °C until LC-MS/MS analysis. Plasma and urine samples (25 μL) were mixed with 100 μL of an internal standard (IS) solution (propranolol 20 ng/mL in methanol) and the mixtures were vortexed for 10 min. After centrifugation (16,000× *g*, 5 min, 4 °C), an aliquot (5 μL) from the supernatant was injected into the LC-MS/MS system. Feces samples (100 mg) were mixed with 400 μL IS solution and the mixtures were vortexed for 10 min. After centrifugation (16,000× *g*, 5 min, 4 °C), an aliquot (5 μL) from the supernatant was injected into the LC-MS/MS system. 

We also investigated the effect of probenecid on the pharmacokinetics of rosmarinic acid. The femoral artery and femoral vein were cannulated with polyethylene tubes (PE-50; Jungdo, Seoul, Korea) while the rats were anesthetized with Zoletil and Rompun (50 mg/kg and 5 mg/kg, respectively, intramuscular injection), and heparinized saline (10 U/mL) was used to prevent blood clotting. Rosmarinic acid was dissolved in saline and probenecid was dissolved in phosphate-buffered saline (PBS). Rosmarinic acid (1 mg/kg) was injected via the femoral vein followed by the intravenous injection of probenecid (10 mg/kg) or a vehicle. Blood samples (approximately 150 µL each) were collected from the femoral artery at 0, 0.033, 0.083, 0.167, 0.25, 0.5, 1, 1.5, 2, 4, and 6 h. After centrifugation of the blood samples at 10,000 g for 1 min, plasma samples (25 µL) were collected and stored at −80 °C until LC-MS/MS analysis. Urine samples were collected for 12 h and aliquots (25 µL) of 10-fold diluted urine samples were stored at −80 °C until LC-MS/MS analysis. Plasma and urine samples were prepared with the method described previously. 

### 2.4. Involvement of Transporters in the Uptake of Rosmarinic Acid 

HEK293-OAT1, -OAT3, -OATP1B1, -OATP1B3, -OCT1, -OCT2, and -NTCP cells and HEK293-mock cells were cultured in DMEM supplemented with 10% FBS and 5 mM non-essential amino acids in a humidified atmosphere of 8% CO_2_ at 37 °C, in DMEM supplemented with 10% FBS, and 5 mM non-essential amino acids. In the case of HEK293-OATP1B1, -OATP1B3, and -NTCP cells, 2 mM sodium butyrate was added to the culture medium. For the experiments, 2 × 10^5^ cells were seeded in poly-D-lysine-coated 24-well plates. After 24 h, the growth medium was discarded and the attached cells were washed with prewarmed HBSS and preincubated for 10 min in HBSS at 37 °C. 

The uptake of 10 μM rosmarinic acid was measured in the absence and presence of representative inhibitors for 5 min at 37 °C. The final concentrations of representative inhibitors were selected based on our previous inhibition results [22]: tetraethylammonium 10 mM (for OCT1 and OCT2), probenecid 50 μM (for OAT1 and OAT3), rifampin 50 μM (for OATP1B1 and OATP1B3), and cyclosporin A 20 μM (for NTCP). After 5 min incubation, cells were then washed twice with 2 mL of ice-cold HBSS immediately after the plates were placed on ice. Subsequently, 150 μL of 80% acetonitrile was added to each sample well and the cell plates were shaken gently for 20 min in a cold room (4 °C). Then 100 μL of each sample was transferred to a clean tube, centrifuged (16,000× *g*, 10 min, 4 °C), and aliquots (75 μL) from the supernatant were transferred to clean tubes and mixed with 350 μL IS solution and the mixtures were vortexed for 10 min. After centrifugation (16,000× *g*, 5 min, 4 °C), an aliquot (5 μL) from the supernatant was injected into the LC-MS/MS system. 

In the concentration-dependent uptake studies, the uptake of various concentrations of rosmarinic acid (0.5–100 μM) was measured in HEK293-OAT1 and -mock cells for 5 min at 37 °C. Samples were prepared with the method described previously. The OAT1-mediated rosmarinic acid uptake was calculated by subtracting the transport rates of the mock cells from those of the HEK293-OAT1 cells. 

### 2.5. Inhibitory Effects of Rosmarinic Acid on the Transport Activities of OATs, OATPs, OCTs, and NTCP

HEK293-OAT1, -OAT3, -OATP1B1, -OATP1B3, -OCT1, -OCT2, and -NTCP cells and HEK293-mock cells were cultured in DMEM supplemented with 10% FBS and 5 mM non-essential amino acids in a humidified atmosphere of 8% CO_2_ at 37 °C, in DMEM supplemented with 10% FBS, and 5 mM non-essential amino acids. In the case of HEK293-OATP1B1, -OATP1B3, and -NTCP cells, 2 mM sodium butyrate was added to the culture medium. For the experiments, 5 × 10^4^ cells were seeded in poly-D-lysine-coated 96-well plates. After 24 h, the growth medium was discarded and the attached cells were washed with prewarmed HBSS and preincubated for 10 min in HBSS at 37 °C.

To investigate the inhibitory effect of rosmarinic acid on OAT1, the uptake of 0.1 μM [^3^H]para-aminohippuric acid in HEK293-OAT1 and -mock cells was measured in the presence of rosmarinic acid (0–500 μM) for 5 min. Similarly, for OAT3 and OATP1B1, the uptake of 0.1 μM [^3^H]estrone-3-sulfate in HEK293-OAT3, -OATP1B1, and -mock cells was measured in the presence of rosmarinic acid (0–500 μM) for 5 min. For OATP1B3, the uptake of 0.1 μM [^3^H]estradiol-17β-D-glucuronide in HEK293-OATP1B3 and -mock cells was measured in the presence of rosmarinic acid (1–500 μM) for 5 min. For OCT1 and OCT2, the uptake of 0.1 μM [^3^H]methyl-4-phenylpyridinium into HEK293-OCT1, -OCT2, and -mock cells was measured in the presence of rosmarinic acid (0–500 μM) for 5 min. For NTCP, the uptake of 0.1 μM [^3^H]taurocholate in HEK293-NTCP and -mock cells was measured in the presence of rosmarinic acid (0–500 μM) for 5 min. After aspirating the incubation medium and washing the cells three times with ice-cold HBSS (200 μL each time), the cells in the plate were lysed with 10% sodium dodecyl sulfate solution (50 μL each). Cell lysates were mixed thoroughly with an Optiphase scintillation cocktail (250 μL each) and the radioactivities in the cocktail mixtures were measured using a liquid scintillation counter. The transporter-mediated uptake of the probe substrate was calculated by subtracting the uptake of probe substrates in HEK293-mock cells from the uptake of the probe substrates in HEK293-OAT1, -OAT3, -OATP1B1, -OATP1B3, -OCT1, -OCT2, and -NTCP cells [23].

### 2.6. Inhibitory Effects of Rosmarinic Acid on the Pharmacokinetics of Furosemide and Valsartan 

We investigated the effect of rosmarinic acid on the pharmacokinetics of furosemide, a substrate for OAT1 and OAT3 [24,25], and valsartan, a substrate for OATP1B1 and OATP1B3 [23,26]. For the pharmacokinetic interaction studies between furosemide and rosmarinic acid, the femoral artery and femoral vein were cannulated with polyethylene tubes (PE-50; Jungdo, Seoul, Korea) while the rats were anesthetized with Zoletil and Rompun (50 mg/kg and 5 mg/kg, respectively, intramuscular injection), and heparinized saline (10 U/mL) was used to prevent blood clotting. Rosmarinic acid was dissolved in saline and furosemide was dissolved in PBS. Furosemide (0.2 mg/kg) was injected via the femoral vein followed by the intravenous injection of rosmarinic acid (10 mg/kg) or a vehicle. Blood samples (approximately 150 µL each) were collected from the femoral artery at 0, 0.033, 0.25, 0.5, 1, 1.5, 2, 4, and 8 h. After centrifugation of the blood samples at 10,000 *g* for 1 min, plasma samples (25 µL) were collected and stored at −80 °C until LC-MS/MS analysis. Urine samples were collected for 12 h and aliquots (25 µL) of 10-fold diluted urine samples were stored at −80 °C until LC-MS/MS analysis.

To investigate OATPs-mediated drug interactions between valsartan and rosmarinic acid, the femoral artery, femoral vein, and bile duct were cannulated with polyethylene tubes (PE-50 or PE-10; Jungdo, Seoul, Korea) while the rats were anesthetized with Zoletil and Rompun (50 mg/kg and 5 mg/kg, respectively, intramuscular injection), and heparinized saline (10 U/mL) was used to prevent blood clotting. Rosmarinic acid was dissolved in saline and valsartan was dissolved in PBS. Valsartan (3 mg/kg) was injected via the femoral vein followed by the intravenous injection of rosmarinic acid (10 mg/kg) or a vehicle. Blood samples (approximately 150 µL each) were collected from the femoral artery at 0, 0.033, 0.083, 0.167, 0.25, 0.5, 1, 1.5, 2, 4, and 6 h. After centrifugation of the blood samples at 10,000 *g* for 1 min, plasma samples (25 µL) were collected and stored at −80 °C until LC-MS/MS analysis. Bile samples were collected for 12 h and aliquots (25 µL) of 10-fold diluted bile samples were stored at −80 °C until LC-MS/MS analysis. 

Plasma, urine, and bile samples (25 μL) were mixed with 100 μL IS solution (propranolol 20 ng/mL in methanol) and the mixtures were vortexed for 10 min. After centrifugation (16,000× *g*, 5 min, 4 °C), an aliquot (5 μL) from the supernatant was injected into the LC-MS/MS system.

### 2.7. LC-MS/MS Analysis

The concentrations of rosmarinic acid and probenecid in the biological samples were analyzed using an Agilent 6430 Triple Quadrupole LC-MS/MS system (Agilent, Wilmington, DE, USA) equipped with an Agilent 1260 HPLC system using a modification of a previously published method [27,28]. The separation was performed on a Polar RP column (2.0 × 150 mm, 5 μm; Phenomenex, Torrance, CA, USA) using a mobile phase consisting of water and methanol (25:75 *v*/*v*) with 0.1% formic acid at a flow rate of 0.2 mL/min. The retention times of rosmarinic acid, probenecid, and propranolol (IS) were 2.1 min, 2.5 min, and 3.2 min, respectively. Quantification was carried out at m/z 359.2 → 161.2 (collision energy (CE) of 10 eV; negative ion mode) for rosmarinic acid, m/z 284.1 → 240.1 (CE of 5 eV; negative ion mode) for probenecid, and m/z 260.0 → 116.1 (CE of 10 eV; positive ion mode) for propranolol. Calibration standards for the measurement of rosmarinic acid in plasma were linear in the range of 0.02–420 μM for plasma, urine, and feces homogenate samples and calibration standards for the measurement of probenecid in plasma were linear in the range of 0.05–700 μM for plasma and urine samples. The intra- and inter-day precision and accuracy had coefficients of variance of less than 15% (Appendix A). The matrix effect and extraction recovery of rosmarinic acid and probenencid from the methanol precipitation method showed coefficients of variance of less than 15% (Appendix A). No significant degradation occurred in the rosmarinic acid and probenecid from the short-term stability (4 h, 25 °C), post-treatment stability (6 °C, 24 h), and freeze-thaw cycle stability (−80 °C/25 °C, 3 Cycles) measurement (Appendix A). 

Furosemide concentrations were analyzed using an Agilent 6470 Triple Quadrupole LC-MS/MS system with a modification of the previously published method [20]. The separation was performed on a Polar RP column (2.0 × 150 mm, 5 μm) using a mobile phase consisting of water and methanol (25:75 *v*/*v*) with 0.1% formic acid at a flow rate of 0.2 mL/min. The retention times of furosemide and propranolol (IS) were 3.6 min and 3.2 min, respectively. Quantification was carried out at m/z 328.9 → 284.8 (CE of 15 eV; negative ion mode) for furosemide and m/z 260.0 → 116.1 (CE of 10 eV; positive ion mode) for propranolol. Calibration standards for the measurement of furosemide in plasma were linear in the range of 2–5000 nM for plasma and urine samples and the intra- and inter-day precision and accuracy had coefficients of variance of less than 15% (Appendix A). The matrix effect and extraction recovery of furosemide from the methanol precipitation method showed coefficients of variance of less than 15% (Appendix A). No significant degradation occurred in the furosemide samples from the short-term stability, post-treatment stability, and freeze-thaw cycle stability measurement (Appendix A). 

Valsartan concentrations were analyzed using an Agilent 6430 Triple Quadrupole LC-MS/MS system utilizing a modification of a previously published method [20,23]. The separation was performed on a Polar RP column (2.0 × 150 mm, 5 μm) using a mobile phase consisting of water and methanol (25:75 *v*/*v*) with 0.1% formic acid at a flow rate of 0.2 mL/min. The retention times of valsartan and propranolol (IS) were 2.6 min and 3.2 min, respectively. Quantification was carried out at m/z 436.1 → 291.0 (CE of 10 eV; positive ion mode) for valsartan and m/z 260.0 → 116.1 (CE of 10 eV; positive ion mode) for propranolol. Calibration standards for the measurement of valsartan in plasma were linear in the range of 0.04–20 μM for plasma and bile samples and the intra- and inter-day precision and accuracy had coefficients of variance of less than 15% (Appendix A). The matrix effect and extraction recovery of valsartan from the methanol precipitation method showed coefficients of variance of less than 15% (Appendix A). No significant degradation occurred in the furosemide samples from the short-term stability, post-treatment stability, and freeze-thaw cycle stability measurement (Appendix A). 

### 2.8. Data Analysis 

The pharmacokinetic parameters of rosmarinic acid, probenecid, furosemide, and valsartan were calculated by a noncompartmental analysis using WinNonlin (version 5.1, Pharsight, Mountain View, CA, USA). T_1/2_ was calculated from the elimination coefficient (K) of plasma concentrations of rosmarinic acid, probenecid, furosemide, and valsartan by the least square regression analysis with a correlation coefficient of over 0.98. AUC_∞_ was calculated from the area under plasma concentrations curve from zero to the last time point (AUC_last_) and the sum of the extrapolated area calculated by C_last_/K and the extrapolated area was less than 5% of AUC_last_ of rosmarinic acid, probenecid, furosemide, and valsartan. The renal clearance of rosmarinic acid (CL_renal_) was estimated by dividing the total amount of rosmarinic acid excreted into urine for 24 h by the plasma AUC of rosmarinic acid.

The transporter-mediated uptakes of rosmarinic acid and probe substrate were calculated by subtracting the uptake of rosmarinic acid and probe substrate in HEK293-mock cells from the uptake of rosmarinic acid and probe substrate in HEK293 cells overexpressing OCT1, OCT2, OAT1, OAT3, OATP1B1, OATP1B3, and NTCP transporters.

The OAT1-mediated uptake rate (*ν*) of rosmarinic acid versus the rosmarinic acid concentration (S) profile was fitted to the Michaelis–Menten equation [ν = V_max_ ([S]/K_m_ + [S])] using WinNonlin [29]. V_max_ indicated the maximum velocity of OAT1-mediated uptake and K_m_ represented the rosmarinic acid concentration that showed half-maximal velocity. The correlation coefficient and standard error of estimates for the K_m_ and V_max_ values of rosmarinic acid were 0.9938 and 2.6%, respectively.

The percentages of inhibition were calculated by the ratio of the amounts of rosmarinic acid in the presence and absence of the inhibitors, and the transporter-mediated uptake rate (*ν*) of probe substrate versus concentrations of inhibitors (I) were fitted to an inhibitory effect equation [ν = E_max_ (1 − [I]/IC_50_ + [I])] using WinNonlin [30,31]. E_max_ indicated the maximum effect and IC_50_ represented the half-maximal inhibitory concentration. The standard error of estimates for the IC_50_ values of rosmarinic acid for OAT1, OAT3, OATP1B1, OATP1B3 were 5.0%, 6.5%, 9.0%, and 10.3%, respectively. 

The statistical significance was assessed by t-test using Statistical Package for the Social Sciences (SPSS Inc., Chicago, IL, USA).

## 3. Results

### 3.1. Pharmacokinetics of Rosmarinic Acid Following Intravenous Injection 

The plasma concentration of rosmarinic acid showed bi-exponential decay following intravenous injection of rosmarinic acid at doses of 1 mg/kg and 10 mg/kg (Figure 2). The pharmacokinetic parameters of rosmarinic acid, such as the dose-normalized AUC (AUC/D), elimination half-life (T_1/2_), systemic clearance (CL), and volume of distribution (Vd), were not statistically different regardless of the intravenous dose (1 mg/kg vs. 10 mg/kg) (Table 1). The recovery of rosmarinic acid for 24 h was 36.4–37.2% from the renal route and 0.80–0.87% from the fecal route (Table 1). This suggested that the major excretion route of rosmarinic acid is urinary excretion rather than biliary excretion. The unrecovered fraction (approximately 62% of the intravenous dose) indicated the metabolic elimination of rosmarinic acid, consistent with previous reports [1,10]. Considering the protein binding of rosmarinic acid (91.4% in rat plasma, our unpublished data) and inulin clearance (6.81 mL/min/kg) [18,32], the glomerular filtration clearance of rosmarinic acid could be calculated as 0.54 mL/min/kg, which is much lower than the CL_renal_ of rosmarinic acid (3.17 mL/min/kg). This suggested that active secretion mediated by the renal transport system can be involved in the urinary excretion of rosmarinic acid. Therefore, further investigation regarding the involvement of the transport system in rosmarinic acid transport has been performed.

### 3.2. Involvement of OAT1 in the Uptake of Rosmarinic Acid 

Involvement of drug transporters in the uptake of rosmarinic acid was measured by comparing the uptake of rosmarinic acid in HEK293 cells overexpressing drug transporters such as OCT1, OCT2, OAT1, OAT3, OATP1B1, OATP1B3, and NTCP with the uptake of rosmarinic acid in HEK293-mock cells. Rosmarinic acid uptake into HEK293-OAT1 cells was increased 115-fold compared with that in HEK293-mock cells and greatly decreased (by 78%) by the presence of probenecid, a representative OAT inhibitor [22]. However, other transporters were not involved in the uptake of rosmarinic acid (Figure 3A).

To characterize the OAT1-mediated rosmarinic acid uptake, we measured the concentration dependence of the rosmarinic acid uptake in HEK293-OAT1 and –mock cells. OAT1-mediated rosmarinic acid uptake was calculated by subtracting the rosmarinic acid uptake in HEK293-mock cells from that in HEK293-OAT1 cells (Figure 3B). Rosmarinic acid uptake was saturated by increasing its concentration and Eadie–Hoftee transformation of OAT1-mediated concentration-dependent uptake of rosmarinic acid revealed the one-site saturation kinetics with a correlation coefficient of 0.9918 (Figure 3C). Therefore, the concentration-dependent rosmarinic acid uptake was fitted to a simple Michaelis–Menten equation, and the kinetic parameters such as K_m_*,* V_max_, and intrinsic clearance (V_max_/K_m_) were calculated as 26.5 μM, 69.0 pmol/min, and 2.60 μL/min, respectively (Figure 3B). 

Then we evaluated the contribution of OAT1 to the pharmacokinetics of rosmarinic acid by investigating the effect of co-administration of probenecid, an OAT1 inhibitor [32], on the in vivo pharmacokinetics and urinary excretion of rosmarinic acid. The plasma concentrations of rosmarinic acid were significantly increased by the co-administration of probenecid (Figure 4A). As a result, the AUC of rosmarinic acid was significantly increased and its CL was decreased by the co-injection of probenecid (Table 2). Because OAT1 is exclusively expressed in the kidney and contributes to the renal excretion of organic anion substrates [33], we also compared the urinary excretion of rosmarinic acid with and without the presence of probenecid. The urinary excretion of rosmarinic acid was significantly decreased by the probenecid co-treatment (Figure 4B), suggesting the role of OAT1 in the renal excretion and pharmacokinetics of rosmarinic acid. Considering the inhibitory coefficient (IC_50_) of probenecid for OAT1 (12.3 μM for human OAT1 and 15.3 μM for rat OAT1) [34] and the plasma concentrations of probenecid in this study (Figure 4C), the plasma concentrations of probenecid exceeded its IC_50_ value for OAT1 for 1.5 h. This suggested that the plasma probenecid concentration is high enough to inhibit OAT1 transport activity, to reduce renal excretion of rosmarinic acid, and to increase its plasma concentration. 

### 3.3. Inhibitory Effects of Rosmarinic Acid on the Transport Activities of OATs, OATPs, OCTs, and NTCP

The inhibitory effects of rosmarinic acid on seven major solute carrier transporters were evaluated using HEK293 cells overexpressing OAT1, OAT3, OATP1B1, OATP1B3, OCT1, OCT2, and NTCP transporters (Figure 5). Rosmarinic acid inhibited organic anion transporters in a concentration-dependent manner with IC_50_ values of 60.6 μM for OAT1, 1.52 μM for OAT3, 74.8 μM for OATP1B1, and 91.3 μM for OATP1B3 (Figure 5A,B). However, rosmarinic acid did not significantly inhibit OCT1, OCT2, and NTCP at concentrations tested up to 500 μM (Figure 5C,D). 

We evaluated the inhibitory effect of rosmarinic acid on OCT1, OAT3, OATP1B1, and OATP1B3 by investigating the pharmacokinetic interaction between rosmarinic acid and the probe substrates for these transporters. Furosemide was selected as a probe substrate for OAT1 and OAT3 [20,25] and valsartan was selected as a probe substrate for OATP1B1 and OATP1B3 [20]. The plasma concentrations of furosemide acid were increased by the co-administration of rosmarinic acid (Figure 6A). As a result, the AUC of furosemide was significantly increased and its CL was decreased by the co-injection of rosmarinic acid (Table 3); urinary excretion of furosemide was significantly decreased by the co-administration of rosmarinic acid (Figure 6B). To determine whether the alteration of furosemide pharmacokinetics was caused by OAT1, OAT3, or both, we analyzed the plasma concentrations of rosmarinic acid (Figure 6C). Considering IC_50_ of rosmarinic acid for OAT1 (IC_50_ 60.6 μM) and OAT3 (IC_50_ 1.52 μM) and the fact that the plasma concentration of rosmarinic acid was 304.6 μM at 2 min and in the range of 55.4 μM–1.25 μM for 2 h (Figure 6C), significant inhibition of OAT3, but not OAT1, by rosmarinic acid could cause the increase in plasma concentrations and the decrease of renal excretion of furosemide in this study. 

However, the co-administration of rosmarinic acid with valsartan did not result in a significant pharmacokinetic drug interaction between rosmarinic acid and valsartan. That is, the plasma concentration and biliary excretion of valsartan were not significantly changed by rosmarinic acid (Figure 7A,B, Table 4). The plasma concentrations of rosmarinic acid were not higher than the IC_50_ values of OATP1B1 and OATP1B3 (Figure 7C), which could account for the lack of pharmacokinetic interaction between rosmarinic acid with valsartan.

## 4. Discussion

Repeated high doses of *Melissa officinalis* extract (containing 120 mg of rosmarinic acid for 16 weeks) have been reported to improve cognitive function in patients with mild-to-moderate Alzheimer’s disease [35]. To determine the acceptable daily intake dose of rosmarinic acid, the pharmacokinetics and tolerability of high dose intake of rosmarinic acid (50 mg/kg in rats and 500 mg in humans) have been investigated [9,13]. Rosmarinic acid showed low oral bioavailability (0.9–1.7% in the oral dose range from 12.5 mg/kg to 50 mg/kg in rats) [13]. In addition, the plasma AUC of rosmarinic acid did not show dose linearity in the dose range of 12.5–50 mg/kg in rats [13] and the dose range of 100–500 mg/kg in humans [9]. The decreased ratio of rosmarinic acid conjugates compared to rosmarinic acid after high dose intake (50 mg/kg in rats and 500 mg in humans) could be attributed to the nonlinearity of rosmarinic acid pharmacokinetics. Taken together, the extensive intestinal metabolism (i.e., glucuronide conjugation of rosmarinic acid) could account for the low oral bioavailability and non-linear pharmacokinetics of rosmarinic acid. 

In the case of herbal medicines with low oral bioavailability and extensive intestinal first-pass metabolism, the therapeutic efficacy could not be properly achieved even though they were administered at their high dose [36]. To overcome this huddle, a lot of formulations with pharmaceutical excipients (e.g., Tween 80, Pluronic P85, and vitamin E-D-α-tocopheryl polyethylene glycol 1000 succinate) that increased the intestinal absorption and reduced the intestinal metabolism of natural herbs with low oral bioavailability such as curcumin, morin, and berberine have been reported [37,38,39]. Apolipoprotein cross-linked polyacrylamide-chitosan-poly (lactide-co-glycolide) nanoparticles of rosmarinic acid showed improvement in the brain permeability and the therapeutic efficacy in rats with experimentally induced Alzheimer’s disease following three intravenous injections of rosmarinic acid formulation [40]. Other formulation studies of rosmarinic acid have been focused on the topical delivery of rosmarinic acid to enhance anti-aging and anti-acne effects [41,42]. Instead, the formulation containing rosmarinic clinically used is the intravenous injection formulation (e.g., Danhong Injection, Shenxiong glucose injection, *Salvia miltiorrhiza* polyphenolic acid solution) for cardiovascular and lung disease [17,27,43]; therefore, the investigation of the pharmacokinetics of rosmarinic acid after intravenous injection would be important. 

Our study results revealed that the AUC values of rosmarinic acid showed dose proportionality after intravenous injection of 1 mg/kg and 10 mg/kg rosmarinic acid; the renal elimination of intact rosmarinic acid was calculated as approximately 35% (Table 1). In contrast, the urinary excretion of intact rosmarinic acid was less than 0.1% following oral administration [2]. The difference in the elimination of rosmarinic acid depending on its administration route also indicated that the different contributions of drug-metabolizing enzymes and transporters to rosmarinic acid pharmacokinetics. OAT1 seemed to be one of the major transporters that modulated the pharmacokinetics, tissue distribution, and renal elimination of rosmarinic acid, as evidenced by the results of the active transport of rosmarinic acid in HEK293 cells overexpressing OAT1 and the decreased renal excretion of rosmarinic acid with co-administration of probenecid, a representative OAT1 inhibitor (Figure 3 and Figure 4). These results also coincide with the significantly higher kidney distribution of rosmarinic acid compared with the distribution in other tissues such as the liver, heart, spleen, lung, and brain [44]. 

In addition, rosmarinic acid inhibited the transport activities of OAT1, OAT3, OATP1B1, and OATP1B3 with IC_50_ values of 60.6 μM, 1.52 μM, 74.8 μM, and 91.3 μM, respectively (Figure 5). Among these inhibitions, the inhibitory effect of rosmarinic acid on OAT3 transport activity caused an in vivo pharmacokinetic drug interaction with intravenous co-administration of furosemide 0.2 mg/kg and rosmarinic acid 10 mg/kg, which resulted in the increased plasma concentration and decreased renal excretion of furosemide (Figure 6). Because the plasma concentrations of rosmarinic acid in humans during treatment with Danhong Injection, Shenxiong glucose injection, or *Salvia miltiorrhiza* polyphenolic acid solution were not available at this time, the clinical relevance of the OAT3-mediated drug interaction of rosmarinic acid needs to be further evaluated. However, the possibility of an OAT3-mediated drug interaction of rosmarinic acid should be kept in mind in the drug development and therapeutics of rosmarinic acid and rosmarinic acid-containing formulae.

## 5. Conclusions

We investigated the pharmacokinetics and excretion of rosmarinic acid following its intravenous injection and reported the involvement of drug transporters in its renal excretion. A comprehensive evaluation of the transport mechanisms and transporter-mediated herb-drug interaction of rosmarinic acid on seven major solute carrier transporters (i.e., OCT1, OCT2, OAT1, OAT3, OATP1B1, OATP1B3, and NTCP) and their role in the in vivo pharmacokinetics of rosmarinic acid was performed. The uptake of rosmarinic acid was greatly increased in HEK293-OAT1 cells, compared with that in -mock cells. Additionally, rosmarinic acid inhibited the transport activities of OAT1, OAT3, OATP1B1, and OATP1B3, with IC_50_ values of 60.6 μM, 1.52 μM, 74.8 μM, and 91.3 μM, respectively. These findings suggested that the OAT1 and OAT3 transporters may be important in the pharmacokinetics and herb–drug interaction of intravenously administered rosmarinic acid, although further evaluation is needed regarding their contribution to its pharmacokinetics in vivo.

## Figures and Tables

**Figure 1 pharmaceutics-13-00083-f001:**
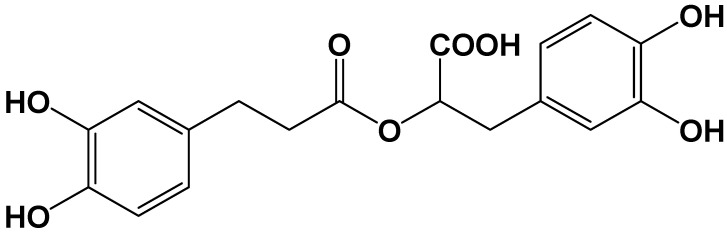
The structure of rosmarinic acid.

**Figure 2 pharmaceutics-13-00083-f002:**
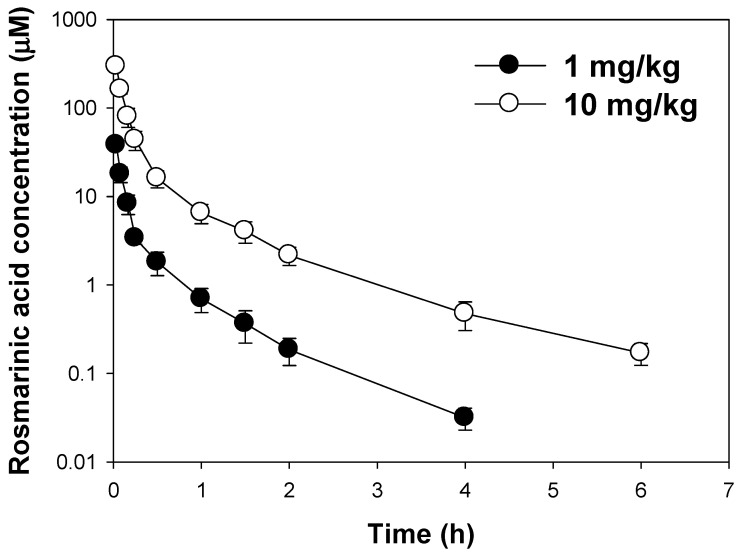
The plasma concentration-time profile of rosmarinic acid following the intravenous injection of rosmarinic acid at doses of 1 mg/kg and 10 mg/kg. Each data point represents the mean ± standard deviation (*n* = 4).

**Figure 3 pharmaceutics-13-00083-f003:**
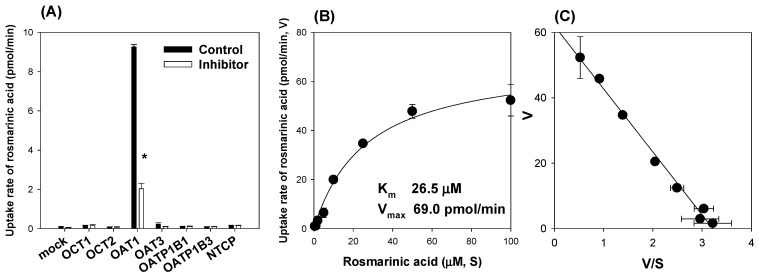
(**A**) The uptake of 10 μM rosmarinic acid into HEK293-mock cells and HEK293 cells overexpressing OCT1, OCT2, OAT1, OAT3, OATP1B1, OATP1B3, and NTCP transporters, was measured for 5 min. The uptake rate of rosmarinic acid in the presence (□) of representative inhibitors such as tetraethylammonium (10 mM) for OCT1 and OCT2, probenecid (50 μM) for OAT1 and OAT3, rifampin (50 μM) for OATP1B1 and OATP1B3, and cyclosporin A (20 μM) for Na^+^/taurocholate cotransporting polypeptide (NTCP) was compared with that in the absence of inhibitors (■). (**B**) The concentration-dependent uptake of rosmarinic acid into HEK293-mock and -OAT1 cells was measured for 5 min in the concentration range of rosmarinic acid (0.5–100 μM). The OAT1-mediated uptake of rosmarinic acid was obtained by subtracting the uptake of rosmarinic acid in HEK293-mock cells from that in HEK293-OAT1 cells. (**C**) Eadie–Hofstee transformation of the OAT1-mediated uptake of rosmarinic acid. V indicated the uptake rate of rosmarinic acid and S indicated rosmarinic acid concentration. Each data point represents the mean ± standard deviation (*n* = 3). * *p* < 0.05 compared with control group.

**Figure 4 pharmaceutics-13-00083-f004:**
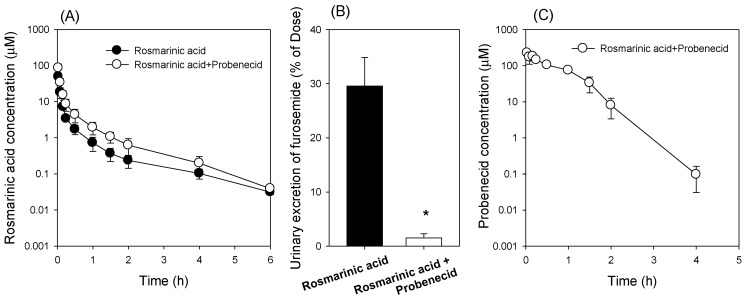
(**A**) The plasma concentration-time profile and (**B**) urinary excretion of rosmarinic acid following the intravenous injection of rosmarinic acid (1 mg/kg) in the absence (●) or presence (○) of probenecid (10 mg/kg). (**C**) Plasma concentration-time profile of probenecid following intravenous injection of probenecid (10 mg/kg) and rosmarinic acid (1 mg/kg). Each data point represents the mean ± standard deviation (*n* = 4). * *p* < 0.05 compared with rosmarinic acid group.

**Figure 5 pharmaceutics-13-00083-f005:**
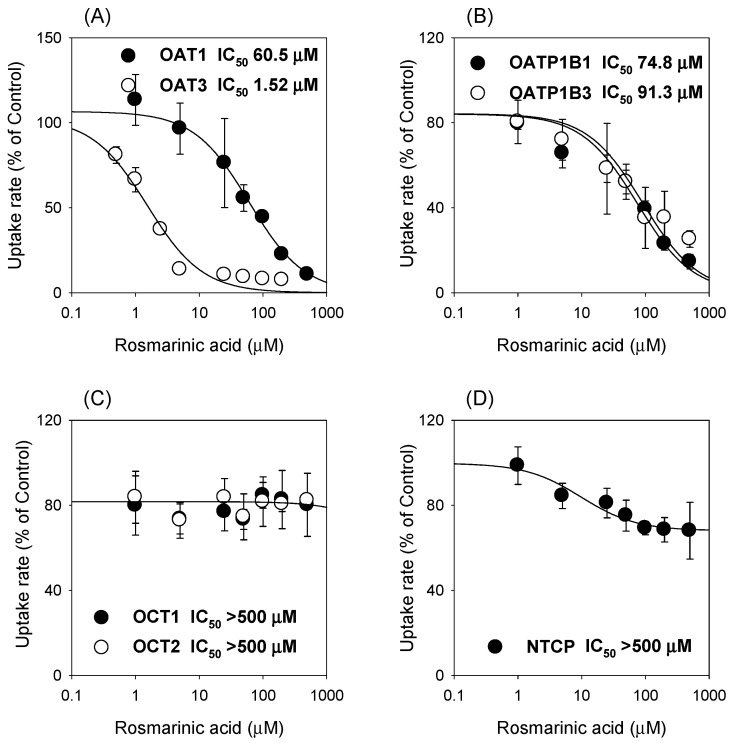
(**A**) The inhibitory effect of rosmarinic acid (1–500 μM) on the OAT1- and OAT3-mediated uptake of 0.1 μM [^3^H]para-aminohippuric acid and 0.1 μM [^3^H]estrone-3-sulfate, a probe substrate for OAT1- and OAT3, respectively. (**B**) The inhibitory effect of rosmarinic acid (1–500 μM) on the OATP1B1- and OATP1B3-mediated uptake of 0.1 μM [^3^H]estrone-3-sulfate and 0.1 μM [^3^H]estradiol-17β-D-glucuronide, a probe substrate for OATP1B1 and OATP1B3, respectively. (**C**) The inhibitory effect of rosmarinic acid (1–500 μM) on the OCT1- and OCT2-mediated uptake of 0.1 μM [^3^H]methyl-4-phenylpyridinium, a probe substrate for OCT1 and OCT2. (**D**) The inhibitory effect of rosmarinic acid (1–500 μM) on the NTCP-mediated uptake of 0.1 μM [^3^H]taurocholate, a probe substrate for NTCP. Each data point represents the mean ± standard deviation (*n* = 3).

**Figure 6 pharmaceutics-13-00083-f006:**
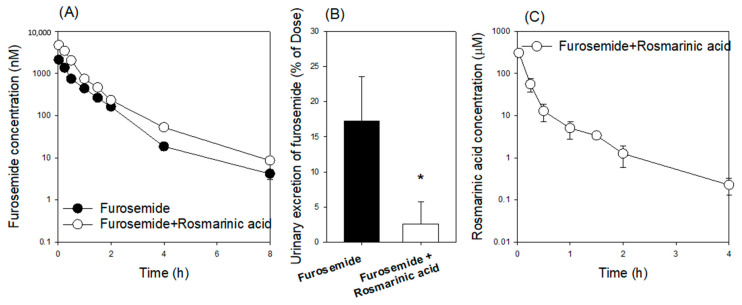
(**A**) The plasma concentration-time profile and (**B**) urinary excretion of furosemide following the intravenous injection of furosemide (0.2 mg/kg) in the absence (●) or presence (○) of rosmarinic acid (10 mg/kg). (**C**) The plasma concentration-time profile of rosmarinic acid following the intravenous injection of furosemide (0.2 mg/kg) and rosmarinic acid (10 mg/kg). Each data point represents the mean ± standard deviation (*n* = 4). * *p* < 0.05 compared with furosemide group.

**Figure 7 pharmaceutics-13-00083-f007:**
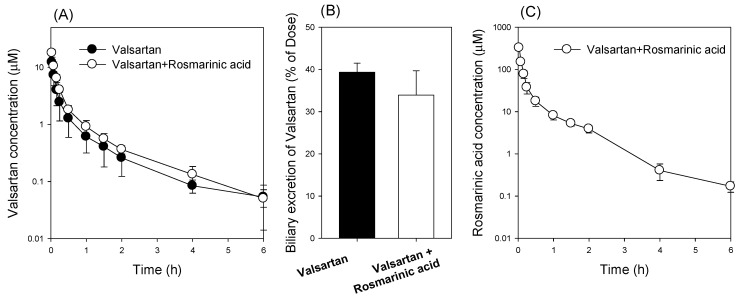
(**A**) The plasma concentration-time profile and (**B**) urinary excretion of valsartan following the intravenous injection of valsartan (3 mg/kg) in the absence (●) or presence (○) of rosmarinic acid (10 mg/kg). (**C**) The plasma concentration-time profile of rosmarinic acid following intravenous injection of valsartan (3 mg/kg) and rosmarinic acid (10 mg/kg). Each data point represents the mean ± standard deviation (*n* = 4).

**Table 1 pharmaceutics-13-00083-t001:** Pharmacokinetic parameters of rosmarinic acid in rats.

Parameters	Dose (IV, mg/kg)	
	1	10
C_0_ (μM)	63.6 ± 5.6	420 ± 98 *
AUC_last_ (μM·h) ^a^	6.35 ± 0.54	64.6 ± 3.2 *
AUC_∞_ (μM·h)	6.36 ± 0.54	65.1 ± 1.8 *
AUC_∞_/Dose (μM·h/mg/kg)	6.36 ± 0.54	6.51 ± 0.18
T_1/2_ (h)	0.89 ± 0.12	1.02 ± 0.07
CL (mL/min/kg)	7.33 ± 0.66	7.12 ± 0.20
Vd (L/kg)	169 ± 15	170 ± 33
X_urine_ (%)	36.4 ± 6.7	37.2 ± 8.3
X_feces_ (%)	0.87 ± 0.17	0.80 ± 0.03
CL_renal_ (mL/min/kg)	2.69 ± 0.47	2.82 ± 0.77

^a^ AUC_last_: AUC from 0 to 4 h for 1 mg/kg dose and AUC from 0 to 6 h for 10 mg/kg dose. * *p* < 0.05 compared with control group. Data expressed as mean ± standard deviation (*n* = 4).

**Table 2 pharmaceutics-13-00083-t002:** Pharmacokinetic parameters of rosmarinic acid and probenecid in rats.

Parameters	Rosmarinic Acid		Probenecid
	Control	Probenecid Treatment	Probenecid Treatment
C_0_ (μM)	98.1 ± 20	162 ± 62	283 ± 59
AUC_last_ (μM·h) ^a^	7.63 ± 0.68	15.6 ± 4.7 *	167 ± 19
AUC_∞_ (μM·h)	7.74 ± 0.73	15.6 ± 4.7 *	168 ± 19
T_1/2_ (h)	1.20 ± 0.51	1.00 ± 0.16	0.30 ± 0.03
CL (mL/h/kg)	6.02 ± 0.56	3.17 ± 0.95 *	3.52 ± 0.39
Vd (mL/kg)	147 ± 44	84.7 ± 19 *	138 ± 5.3

^a^ AUC_last_: AUC from 0 to 6 h for rosmarinic acid and AUC from 0 to 4 h for probenecid. * *p* < 0.05 compared with control group. Data expressed as mean ± standard deviation (*n* = 4).

**Table 3 pharmaceutics-13-00083-t003:** Pharmacokinetic parameters of furosemide and rosmarinic acid in rats.

Parameters	Furosemide		Rosmarinic Acid
	Control	+Rosmarinic Acid	+Rosmarinic Acid
C_0_ (μM)	2.25 ± 0.21	4.89 ± 0.39 *	397 ± 23
AUC_last_ (μM·h) ^a^	1.51 ± 0.11	3.29 ± 0.05 *	68.3 ± 11
AUC_∞_ (μM·h)	1.52 ± 0.11	3.30 ± 0.06 *	68.6 ± 11
T_1/2_ (h)	0.97 ± 0.04	1.28 ± 0.15	0.68 ± 0.1
CL (mL/h/kg)	6.68 ± 0.47	3.06 ± 0.05 *	8.67 ± 1.3
Vd (mL/kg)	353 ± 23	148 ± 6.6 *	118 ± 11

^a^ AUC_last_: AUC from 0 to 8 h for furosemide and AUC from 0 to 4 h for rosmarinic acid. * *p* < 0.05 compared with control group. Data expressed as mean ± standard deviation (*n* = 4).

**Table 4 pharmaceutics-13-00083-t004:** Pharmacokinetic parameters of valsartan and rosmarinic acid in rats.

Parameters	Valsartan		Rosmarinic Acid
	Control	+Rosmarinic Acid	+Rosmarinic Acid
C_0_ (μM)	17.8 ± 1.3	25.8 ± 4.5 *	542 ± 56
AUC_last_ (μM·h) ^a^	3.57 ± 1.3	5.27 ± 0.81	64.4 ± 6.9
AUC_∞_ (μM·h)	3.73 ± 1.2	5.41 ± 0.83	64.7 ± 7.0
T_1/2_ (h)	2.03 ± 1.4	1.40 ± 0.24	0.88 ± 0.17
CL (mL/h/kg)	33.3 ± 12	21.7 ± 3.7	9.12 ± 0.98
Vd (mL/kg)	2696 ± 2012	1183 ±147	250 ± 34

^a^ AUC_last_: AUC from 0 to 6 h for valsartan and rosmarinic acid. * *p* < 0.05 compared with control group. Data expressed as mean ± standard deviation (*n* = 4).

## Data Availability

The data presented in this study are available upon request.

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
