# Peer review of "Involvement of Organic Anion Transporters in the Pharmacokinetics and Drug Interaction of Rosmarinic Acid"

_pharmaceutics, 2021, doi:10.3390/pharmaceutics13010083_

Round 1

Reviewer 1 Report

In this study, the plasma concentrations of the parent rosmarinic acid and conjugated rosmarinic  57 acid were about 20 nM and 1200 nM, respectively, at 1 h after oral intake of the Perilla frutescens  58 extract

Obs: "In this work" the conjugate does not appear.
The wording needs to be changed.

Obs, Which is
rosmarinic acid  solubility?  Partition coefficient ? The main problem of the paper is that,
going to very low concentrations,
the estimates regarding
the elimination constant are no longer plausible.
Elimination is no longer bi-exponential but tri-exponential. The extrapolated areas
do not differ much from those up to 6 hours.
As a result, it would be sufficient to do the analysis
only for up to 6 hours.

Reviewer 2 Report

The whole work is interesting.

It lacks some minor aspects such as :
1/-the evaluation of the matrix effect in analytical methods and details of the performance of the assay;
2/-the criteria for the choice of models describing their data;

3/-the graphical detail on residuals or prediction deviations when the model is selected.

Round 2

Reviewer 1 Report

  In this form, the work seems to me correct and complete